# The Evolving Role of Caveolin-1: A Critical Regulator of Extracellular Vesicles

**DOI:** 10.3390/medsci8040046

**Published:** 2020-11-04

**Authors:** Kareemah Ni, Chenghao Wang, Jonathan M Carnino, Yang Jin

**Affiliations:** Division of Pulmonary and Critical Care Medicine, Department of Medicine, Boston University Medical Campus, 72 E Concord St. R304. Boston, MA 02118, USA; kani@bu.edu (K.N.); wangchenghao@cau.edu.cn (C.W.); jcarnino@bu.edu (J.M.C.)

**Keywords:** caveolin-1, extracellular vesicle (EVs), EV cargo

## Abstract

Emerging evidence suggests that extracellular vesicles (EVs) play an essential role in mediating intercellular communication and inter-organ crosstalk both at normal physiological conditions and in the pathogenesis of human diseases. EV cargos are made up of a broad spectrum of molecules including lipids, proteins, and nucleic acids such as DNA, RNA, and microRNAs. The complex EV cargo composition is cell type-specific. A dynamic change in EV cargos occurs along with extracellular stimuli and a change in the pathophysiological status of the host. Currently, the underlying mechanisms by which EVs are formed and EV cargos are selected in the absence and presence of noxious stimuli and pathogens remain incompletely explored. The term EVs refers to a heterogeneous group of vesicles generated via different mechanisms. Some EVs are formed via direct membrane budding, while the others are produced through multivesicular bodies (MVBs) or during apoptosis. Despite the complexity of EV formation and EV cargo selection, recent studies suggest that caveolin-1, a well-known structural protein of caveolae, regulates the formation and cargo selection of some EVs, such as microvesicles (MVs). In this article, we will review the current understanding of this emerging and novel role of cav-1.

## 1. Introduction

### 1.1. EV History, Nomenclature, and Categories

Although initially described in the 1940s, extracellular vesicles (EVs) were not well characterized until only recently. These nano-sized vesicles were first described as platelet-derived particles in 1946 [1]. In the past several decades, there have been many milestones in EV research. In 1983, two reports published in the Journal of Cell Biology [2] and Cell [3] described ~50 nm-size vesicles released from maturing blood reticulocytes into the extracellular space. These nano-vesicles were then named “exosomes” and later the term “exosome complex” was used. In the 2000s, researchers found that all cells are capable of secreting vesicles, however, extracellular vesicles (EVs) were named differently in previous literature, such as microparticles (MPs), nanoparticles, and exosomes. In 2006–2007, EV cargos were identified. Further discovery has shown that EVs contain nucleotides, including RNA, microRNA (miRNA), and DNAs. Since then, researchers have had an increased interest in studying EVs. So far, EVs have been isolated from most cell types and bodily fluids including saliva, urine, nasal and alveolar bronchial lavage fluid (BALF), amniotic fluid, breast milk, plasma, serum, and seminal fluid [4,5,6,7,8].

Extracellular vesicles (EVs) are a group of heterogeneous vesicles with a broad size range of 20–5000 nm and are released from almost all types of cells [9]. Based on current guidelines issued from the International Society of Extracellular Vesicles (ISEV), the nomenclature of EVs is categorized by the size or origin of the vesicles, such as small, medium, and large EVs, or epithelial EVs and endothelial EVs [10]. However, many researchers prefer naming EVs using a former category, i.e., exosomes (EXOs), microvesicles (MVs), and apoptotic bodies (ABs). This three-group category is defined based on not only the size of EVs, but also the mechanism of generation and their cell surface markers (Figure 1). Generally, ABs are the largest EVs with the broadest size range, usually from 500 nm to 1 µm, similar to the size range of platelets. However, ABs are often generated by actively dying cells. As more interest focuses on the smaller vesicles generated from live cells, in the narrower sense, EVs often only refer to MVs and exosomes. In this article, we use EVs as the generic term to avoid confusion. In this review, EVs refer to the previous MVs and exosomes, a narrower sense of the cell-secreted vesicles. Previously named MVs are the middle-size EVs ranging from 200 to 1000 nm and resulting from direct plasma budding. Previously named EXOs range from 20 to 200 nm in size and are generated through multivesicular bodies (MVBs) in the endosomal pathway [9]. These two subpopulations of EVs have different particle sizes with some overlap, a distinct biogenesis pathway, and specific surface markers (Figure 1 and Table 1).

### 1.2. EV Biogenesis ++

With the complex and various naming systems for EVs, we hereby use the term EVs to discuss extracellular vesicles as a whole. Furthermore, EVs in this article only refer to live cell-generated EVs, and not the apoptotic bodies resulting from dying cells. Despite their well-known heterogeneity, EVs share some common characteristics. First, all EVs have a lipid bilayer membrane similar to the plasma membrane of cells. Second, they carry a diverse pattern of cargos which indicate their unique pathways during biogenesis.

The biogenesis of the smallest EVs (formerly called EXOs) is the most complex one. Initially, inward protrusion of the late endosomal membrane results in continuous accumulation of intraluminal vesicles (ILVs) in multivesicular bodies (MVBs), a type of late endosome [11,12,13,14]. MVBs fuse with the plasma membrane or lysosome membrane. If MVBs fuse with lysosomes, the released contents from MVBs are subject to degradation. If MVBs fuse with the plasma membrane, the ILVs released from MVBs will be secreted out of the cells. The endosomal sorting complex is usually required for transport (ESCRT) complexes to regulate the generation of ILVs and the formation of MVBs. Furthermore, tetraspanins, including CD81, CD63, and CD9, are highly enriched in EXOs. Ubiquitinated cargos are sorted into the late endosomes facilitated by ESCRT complexes. Alix, TSG101, syntenin, and syndecans have been reported to participate in EV biogenesis [15,16,17].

Small to medium-sized EVs (formerly called MVs), on the other hand, shed from the plasma membrane via direct budding [12]. The process involves calcium accumulation (Ca^2+^), imbalance of the phospholipid orientation, as well as disruption of the equilibrium of phospholipids, phosphatidylserine (PS), and phosphatidylethanolamine (PE). The symmetry of phospholipids is maintained by enzyme translocases such as scramblases, flippases, and floppases. Accumulation of Ca^2+^ can deactivate flippase, but activates floppase and scramblase. All of these alterations of translocases lead to cytoskeletal imbalance and subsequently result in lipid bilayer outward budding of the plasma membrane [18,19,20].

The formation of the largest EVs (formerly called apoptotic bodies) is relatively simpler. Apoptosis leads to disassembly of dying cells and fragmentation of the cellular components into distinct vesicles, formerly named ABs or currently called the largest EVs [21].

### 1.3. EV Cargos, EV Production, and Their Potential Regulators

Since 2006, many different molecules have been identified as EV cargos, including but not limited to, proteins, RNAs, lipids, and metabolites [22]. Accumulating evidence suggests that a particular group of proteins or RNAs are enriched in EVs. Furthermore, the EV cargo protein or RNA profiles are altered according to the status of the “parent cells” and the noxious stimuli that the “parent cells” have been exposed to [10]. For example, in lung epithelial cells, exposure to hyperoxia, acid, or bacteria dramatically alters the EV cargo miRNA profiles. A repertoire of miRNAs known to promote pro-inflammatory activities are increased robustly in lung epithelium-derived EVs [23]. Similar findings have been found with EV cargo proteins, for example, “parent cells” with exposure to noxious stimuli give rise to EVs with certain sets of proteins whose levels are elevated or reduced [10]. Current evidence suggests the existence of specific sorting mechanisms that arrange the selective packaging of EV cargos. Despite that the underlying mechanisms of EV cargo sorting remain obscure, a few hypotheses of the molecular machinery involved in this process have been proposed. One of the most studied groups of proteins that regulate EV production and cargo sorting is the endosomal sorting complexes required for transport (ESCRT) proteins. The sorting of protein cargos into EVs has been reported to follow specific mechanisms dependent on the ESCRT machinery, tetraspanins, and lipids. As mentioned above, ESCRT protein complex regulates EV generation through the multivesicular body (MVB) pathway. There are five members in the ESCRT machinery: ESCRT-0, -I, -II, -III, and Vps4 complex [24]. The EVs generated via ESCRT-mediated pathways often have smaller sizes, carry distinct marker proteins such as Alix and TSG101, and are also known as exosomes. The selective sorting of protein cargos into EVs has been reported to be regulated by post-translational modifications (PTMs) [25], such as SUMOylation, ubiquitination, phosphorylation, and palmitoylation of proteins. RNA molecules are often packaged into the EVs along with RNA-binding proteins. Since the small EVs (sEVs, previously called exosomes) are generated through the endoplasmic reticulum, the Golgi apparatus, and MVBs [25], proteins and RNAs are more likely to be packaged via ESCRT-independent pathways. However, MVBs can also be produced in an ESCRT-independent pathway. Therefore, cargos in sEVs may also be selected under other regulators in an ESCRT-independent manner [25].

On the other hand, many medium-sized EVs are produced via direct plasma budding as stated above, and therefore are comprised of a lipid bilayer that carries a variety of cell surface proteins and nucleic acids [26].

Despite detailed mechanisms remaining unclear, MV cargos are sorted in a tightly regulated manner rather than randomly chosen. So far, a variety of proteins have been reported to be specifically selected under certain conditions. For instance, Rab GTPases have been reported to regulate MV secretion by neuroblastoma [27]. Other proteins that are found abundant in MVs include cytoskeletal proteins, such as actins, and lipid raft proteins. Besides proteins, a variety of mRNAs and non-coding RNAs have been identified in EVs [28]. Accumulating evidence indicates that nucleic acid contents in MVs may have multiple functions including the signal exchange among cells [23]. In this report, we will focus on the role of lipid raft proteins and their role in EV production and cargo sorting, particularly in membrane budding-generated EVs, formerly named MVs.

Lipid rafts are specialized structures on the plasma membrane [29]. Cholesterol-rich lipid rafts have been demonstrated to play an essential role in EV release from platelets [30]. Lipid rafts are also often called lipid microdomains, which are enriched in cholesterol, glycosphingolipids, and glycosyl-phosphatidylinositol GPI-anchored proteins [31]. Lipid rafts are insoluble in detergent and are involved in the compartmentalization of cell surface molecules and endocytosis of extracellular molecules. There are two types of lipid rafts: planar lipid rafts and caveolae [29] (Figure 2A). Here, we will focus on caveola and its primary component protein, caveolin-1. Caveola is a specific type of lipid raft which has been extensively studied.

## 2. Caveolin-1

### 2.1. Brief Introduction of Caveolae and Caveolins

Caveola, a term derived from Latin, meaning “little caves”, is a cellular structure with a flask-shaped invagination on the plasma membrane and can be found in many types of cells [32]. First described in 1955, caveolae are roughly 50–100 nm in size and have been described as a special type of lipid raft [33]. Caveolae are mainly composed of lipids and proteins such as sphingolipids, cholesterol, and caveolins [32,33]. Caveolins are a group of membrane proteins located both inside and outside of caveolae. The family of caveolin proteins has three members including caveolin-1 (cav-1), caveolin-2 (cav-2), and caveolin-3 (cav-3), which are coded by genes CAV1, CAV2, and CAV3, respectively [34]. Cav-1 and cav-2 are ubiquitously expressed in many different tissues except striated muscle and cardiac muscle cells. Cav-3, also known as M-caveolin, is the muscle-specific form of caveolin [35,36]. Cav-1 and cav-3 are required for caveolae formation in non-muscle cells or muscle cells, respectively [37]. Although cav-2 is thought to be heavily involved in caveolae formation along with cav-1 and may have antagonistic activities against cav-1 [38], the role of cav-2 remains unclear. Among cav-1, cav-2, and cav-3, cav-1 is the most studied and characterized one. In this manuscript, we will focus on the newly identified roles of cav-1 in EV research.

Cav-1 is expressed ubiquitously and differentially among different tissues [39]. Inside cells, cav-1 is found mainly in the membranes of the mitochondria, nucleus, Golgi complex, and endoplasmic reticulum, as well as on the plasma membrane [40]. Based on current information published on “Genecards”, the lung is one of the organs in which cav-1 is expressed most prominently [41]. In the lung epithelium, cav-1 is almost exclusively expressed in alveolar epithelial type I cells (ATI) rather than type II cells (ATII) [42]. Therefore, cav-1 is often used as one of the ATI cell markers in lung research.

Cav-1 is known to have two isoforms: cav-1α and cav-1β. Although recent reports suggest that cav-1α and cav-1β are differentially involved in fetal lung and blood vessel development [43], the tissue and cell distribution of these two forms have not been completely explored.

### 2.2. Caveolin-1 Structure and Isoforms

The cav-1 structure is tightly related to its function and involvement in the regulation of EV secretion and EV cargo selection. Therefore, we will describe the cav-1 molecular structure in detail here. Many of the following mentioned motifs and domains, along with their post-transcriptional modifications, are key factors dictating EV cargo selection.

Cav-1 is a 20.5 kDa integral membrane protein with a hairpin-like conformation (Figure 2) and four major structural domains, including the N-terminal domain (NTD, residues 1–81), the scaffolding domain (CSD, residues 82–101), the intramembrane domain (IMD, residues 102–134), and the C-terminal domain (CTD, residues 135–178) [44]. In cav-1, the two N- and C-terminal tails are facing the cytoplasm and are separated by a hydrophobic intramembrane loop structure [45].

The C-terminal domain (135–178 amino acids) (CTD) is reported to facilitate the relocation of cav-1 from the Golgi apparatus to the plasma membrane and subsequent attachment to the membrane by oligomers. Cav-1 oligomers are the key structures that stabilize caveolae and interact with signaling molecules [46,47,48,49].

The intramembrane domain (102–134 amino acids) (IMD) refers to the membrane-inserting segment of cav-1. IMDs are inserted into the plasma membrane in a cholesterol-dependent manner. The IMD forms an α-helical hairpin that does not completely cross the plasma membrane [50]. The membrane curvature of caveolae probably leads to the formation of a hairpin-shaped IMD [51]. For example, NMR studies show a “helix-break-helix” conformation in residues 96–136 of cav-1 [51,52]. Furthermore, in the IMD, Gly108, Gly116, and Pro110 have been shown as essential residues for the formation of this hairpin-shaped structure of cav-1. Even though the IMD plays a key role in the hairpin structure formation, it is not required for membrane binding. On the other hand, the two domains (amino acids 82–101 and 135–150) near the IMD are more important in mediating the binding of cav-1 to membranes [50,52].

The scaffolding domain (82–101 amino acids) (CSD) is a homologous domain that exists in both cav-1 and cav-3 [44] and binds to caveolin-binding motifs (CBM). CBMs are found in many proteins, such as in eNOS, PKA, G-protein, and EGFR, which have conserved motifs enriched with aromatic residues. In the CSD, cav-1 residues 92–95 (FTVT) are required for cav-1 to bind with G-protein α-subunits. The cav-1 binding proteins have conserved aromatic-rich motifs with ΦXΦ XXXXΦ, ΦXXXXΦXXΦ, and ΦXΦXXXXΦXXΦ (Φ = aromatic residue, X = any amino acid) [44]. Similar to many other membrane proteins, cav-1 also contains a motif known as the cholesterol recognition/interaction amino acid consensus (CRAC) sequence, which has a preferential association with cholesterol [53]. The CRAC motif, which participates in the recognition and interaction of cholesterol in caveolae [50] and the aggregation of cav-1, is found on the C-terminal side of the CSD at residues 94–101 (VTKYWGYR) of cav-1. In the CRAC motif of cav-1, V94, Y97, and R101 are essential residues in the interactions between the cav-1 CSD and cholesterol. In the N-terminal side of the CSD, cav-1 residues 84–94 can form a β-sheet hairpin required for the self-oligomerization of cav-1 [50].

The N-terminal domain (NTD) refers to cav-1 residues 1–81 that form a soluble segment. In the NTD, several essential phosphorylation sites play crucial roles in mediating protein–protein interactions. At physiological pH, the secondary structure of the cav-1 NTD is primarily unstructured with random coils. Although the NTD does not appear to interact with the plasma membrane, segments of α-helices and β-strands in the NTD of cav-1 have been predicted to be present [54,55,56].

The difference between cav-1α and cav-1β, the isoforms of cav-1, is in the N-terminal tail, which is a result of either alternative translation initiation sites [57] or mRNA alternative splicing [58]. Interestingly, a well-known cav-1 phosphorylation site (cav-1 tyrosine 14 (Y14)) is only located in cav-1α [59]. Current findings suggest that cav-1α and cav-1β carry distinct roles in embryo development associated with actin cytoskeletal organization and vascular formation. However, the functions of cav-1α and cav-1β do not overlap [60]. Nohe et al. reported that epidermal growth factor treatment results in rearrangement of cav-1α and cav-1β on the cell membrane and dynamic changes in the cav-1 isoform ratio [61]. Cav-1α and cav-1β are also differently expressed in different cells, tissues, and organs at the different developmental stages. For example, cav-1α is found in adult type I lung epithelial cells, but not detected in fetal or neonatal lung epithelium [43].

### 2.3. Caveolin-1 Expression in EVs, Role in EV Generation, and EV Uptake by Recipient Cells

Cav-1 has been reported to regulate endocytosis, exocytosis, signaling transduction, form a platform for signaling molecule assembly, and regulate cell membrane curvature. Recently, emerging evidence has uncovered a novel role of cav-1, regulation of EV formation, and EV cargo sorting.

Cav-1, with its carboxy- and amino-terminal facing the cytoplasm, is an integral membrane protein found in endoplasmic reticulum membranes (Figure 2). Cav-1 has been identified in the secretory pathway of exocrine cells [62], such as pancreatic acinar cells. Cav-1 has also been identified in many other EVs, including a variety of malignant cell-derived EVs [63,64,65].

Cav-1-containing cells may also use EVs as vehicles to remove intracellular cav-1 and secrete them in an EV cargo manner [66]. Furthermore, after noxious stimulation, cav-1 levels in EVs have been reported to robustly increase in a variety of cells [67,68]. These findings raise the question of whether cav-1 facilitates EV secretion. Despite no official reports, current data suggest that cav-1 promotes EV production and secretion, whereas the deletion of cav-1 significantly decreases EV secretion (unpublished data from Dr. Jin’s group, Boston University, Boston, MA, USA).

On the other hand, cav-1 certainly has been proven to regulate EV endocytosis. Moreover, cav-1-dependent endocytosis can internalize EVs, which subsequently regulates the proliferation, migration, invasion, and metastasis of malignant cells [69,70,71,72,73].

EVs can be taken up by targeted recipient cells via phagocytosis, macropinocytosis, fusion with the plasma membrane, clathrin-mediated endocytosis, and caveolin-mediated endocytosis [70,73]. Cav-1 knockout cells have been shown to take up more exosomes than control cells do. Rescuing with cav-1 over-expression plasmids in cav-1 knockout cells reduces exosome uptake [74]. The regulation of exosome uptake by cav-1 is independent of clathrin-mediated uptake or macropinocytosis. Although it is difficult to demonstrate whether the exosomes mentioned above are indeed taken into the recipient cells or simply attach to the cell surface, some specific signaling pathways that regulate exosome uptake have been identified and reported. For example, Cav-1 suppresses p-ERK1/2, which subsequently suppresses p-HSP27 and ultimately reduces exosome uptake [74]. This result suggests that regardless of EV intake or EV attachment to the recipient cells, cav-1 may play a regulatory role. More recently, Yue et al. reported that cav-1-mediated EV intake has been identified in neurons to regulate neuron apoptosis and this uptake occurs through the Cav-1-dependent endocytosis pathway [71]. The authors concluded that cav-1 mediates neuronal protection through mediating EV intake in ischemic injury [71].

### 2.4. Role of Cav-1 in EV Cargo Sorting and Transportation

EV cargo proteins and nucleotides are selected specifically rather than randomly. Cav-1 has been reported to participate in the sorting of EV cargos. Campos et al. reported that EV cargo cav-1 promotes migration and invasiveness of breast cancer cells by enhancing specific EV cargo proteins, including specific cell adhesion-related proteins, such as Cyr61, S100A9, and tenascin [75]. In mesenchymal stem cells (MSCs), Kou et al. illustrated that the Fas/Fap1/Cav-1 axis regulates SNAP25/VAMP5-associated interleukin 1 receptor antagonist (IL-1RA)-secreting small EVs [76]. In their models, cav-1 interacts with Fas and SNAP25, and protein tyrosine phosphatase Fap-1 interacts with both Fas and cav-1. Together, these proteins form a complex that controls the secretion of IL-1RA-containing small EVs.

EVs carry a repertoire of heterogeneous cellular cargos, which not only includes proteins and lipids but also DNA molecules, RNA molecules, and small RNA molecules including microRNA (miRNA), tRNA, and rRNA. These diverse EV cargos have been reported to mediate a variety of biological functions and serve as potential biomarkers [77]. Most EV cargo RNAs tend to be around 200 nucleotides or smaller [78]. More than 1000 miRNAs have been reported [79]. Argonaute 2 (Ago2), lipoproteins, and several heterogeneous nuclear ribonucleoproteins (hnRNPs) have also been reported in EVs, suggesting a potential role of these RNA-binding proteins in facilitating RNA molecule transfer from the parent cells to EVs [80,81]. The EV cargo RNA molecules, including EV cargo miRNAs, are different from the RNA contents in their “mother” cells, indicating a tightly controlled regulatory machinery that sorts the distinct RNA molecules into the EVs. Cav-1 has been reported to serve as one of the components in this regulatory complex, and facilitates the selective loading of miRNAs into EVs in response to specific noxious stimuli [82]. Cav-1 is robustly detected in miRNA-rich EVs compared to miRNA-scarce EVs.

EV cargo miRNAs are also often transported into EVs via their bound proteins, such as hnRNPs and Ago2. HnRNPA2B1 is one of the isoforms in the A/B subfamily of RNA-binding hnRNPs. HnRNPA2B1 is present in the nucleus and shuttled to the cytoplasm and EVs [83]. Furthermore, hnRNPA2B1 has been reported to selectively interact with a specific repertoire of miRNAs, and subsequently transport these miRNAs into EVs [83]. Lee et al. described that cav-1 is present and augmented in lung epithelial cell-derived EVs after exposure to hyperoxia, a form of oxidative stress that often occurs in the respiratory tract. They found that cav-1 facilitates hnRNPA2B1 transportation into these epithelial EVs, and subsequently facilitates the secretion of hnRNPA2B1-bound miRNAs into EVs [67].

Post-translational modifications (PTMs) of cav-1 induced by noxious stimuli play a key role in the process of cav-1-mediated EV cargo sorting. For example, in the presence of oxidative stress, the tyrosine 14 (Y-14) of cav-1 is phosphorylated. The cav-1 Y-14 phosphorylation (pY14) brings a change of cav-1 conformation and exposes the cav-1 scaffolding domain (CSD) to hnRNPA2B1. The CSD and hnRNPA2B1 RGG domain subsequently come closer and interact with each other [67]. Lee et al. further reported that cav-1 tyrosine 97 (Y97) and phenylalanine 99 (F99) in the CSD function as docking sites to promote cav-1/hnRNPA2B1 binding [67]. Ostermeyer et al. also reported that cav-1 Y97, tryptophan 98 (W98), F99, and tyrosine 100 (Y100) are all responsible for the trafficking of cav-1 to the plasma membrane [84]. However, no clear evidence indicates that W98 and Y100 participate in the sorting of miRNAs via hnRNPA2B1. Whether W98 and Y100 interact with other hnRNPs or RNA-binding proteins requires further investigation. On the other hand, the arginine-glycine-glycine repeat (RGG) of hnRNPA2B1 undergoes O-glcNAcylation in the presence of oxidative stress. O-GlcNAcylation of serine 73 (S73) and serine 90 (S90) in the RNA-binding region of hnRNPA2B1 plays an essential role in hnRNPA2B1–miRNA interactions (Figure 3).

## 3. Future Directions and Questions to be Answered

Despite emerging evidence suggesting that cav-1 is actively involved in EV production, secretion, and EV cargo selection, many more questions remain to be answered. For example, is cav-1 required for small EV (formerly named exosomes) generation? Given that cav-1 is a primary component of lipid rafts, it is clearly understood that cav-1 participates in medium/large EVs production when they are generated via direct plasma membrane budding. However, further studies are needed to illustrate whether cav-1 participates in endosome MVB-dependent pathways of small EVs (exosomes) generation, and if so, how and when does cav-1 participate in EV production? Additionally, whether cav-1 participates in the cargo selection in this scenario also requires further investigations.

Second, it is well known that there are two isoforms of cav-1. Therefore, the next question will be whether the different isoforms of cav-1 play differential roles in regulating EV production and/or EV cargo sorting?

Lastly, although less significant, EV production and EV cargo selection remain unchanged when cav-1 is deleted. Therefore, the third question is: which protein replaces the role of cav-1 after cav-1 deletion?

## 4. Conclusions

In summary, cav-1, a previously well-reported lipid raft protein, now has an emerging role in regulating EV generation, secretion, and EV cargo selection.

## Figures and Tables

**Figure 1 medsci-08-00046-f001:**
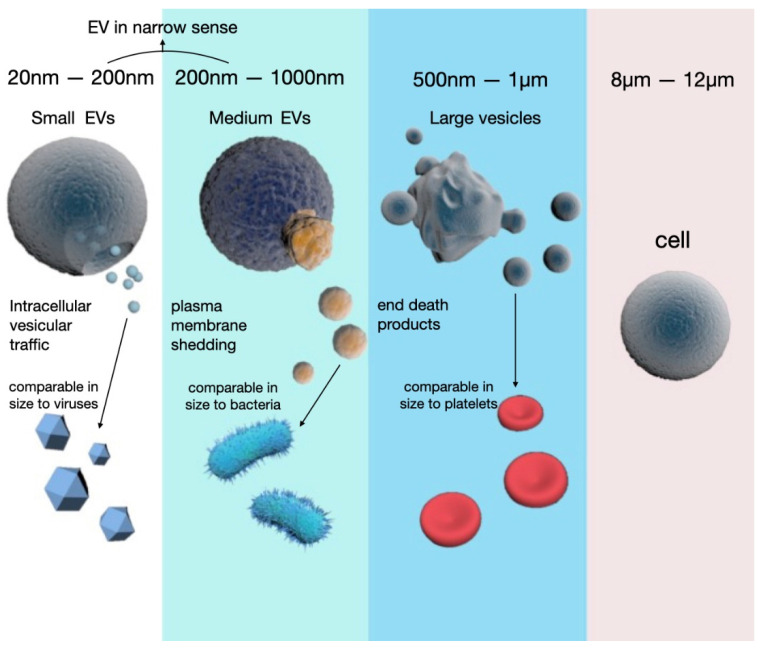
Schematic diagram of extracellular vesicle size and biogenesis. Extracellular vesicles (EVs) refer to a group of heterogeneous vesicles ranging from 20 nm to 5 μm. Small vesicles ranging from 20 to 200 nm used to be named exosomes, while small to medium-size vesicles ranging from 200 to 500 nm used to be called microvesicles (MVs). The largest EVs, which used to be called apoptotic bodies (ABs), are often generated from dying cells. The biogenesis of these EVs has distinct pathways. As illustrated here, small EVs are often generated via a long journey including endosomes, ER/Golgi, and multivesicular bodies (MVBs). On the other hand, the other larger EVs can be produced via direct budding from the plasma membrane. It is commonly facilitated by lipid raft proteins as discussed in this review. The largest-size EVs, or previously referred to as ABs, are broken down from apoptotic cells.

**Figure 2 medsci-08-00046-f002:**
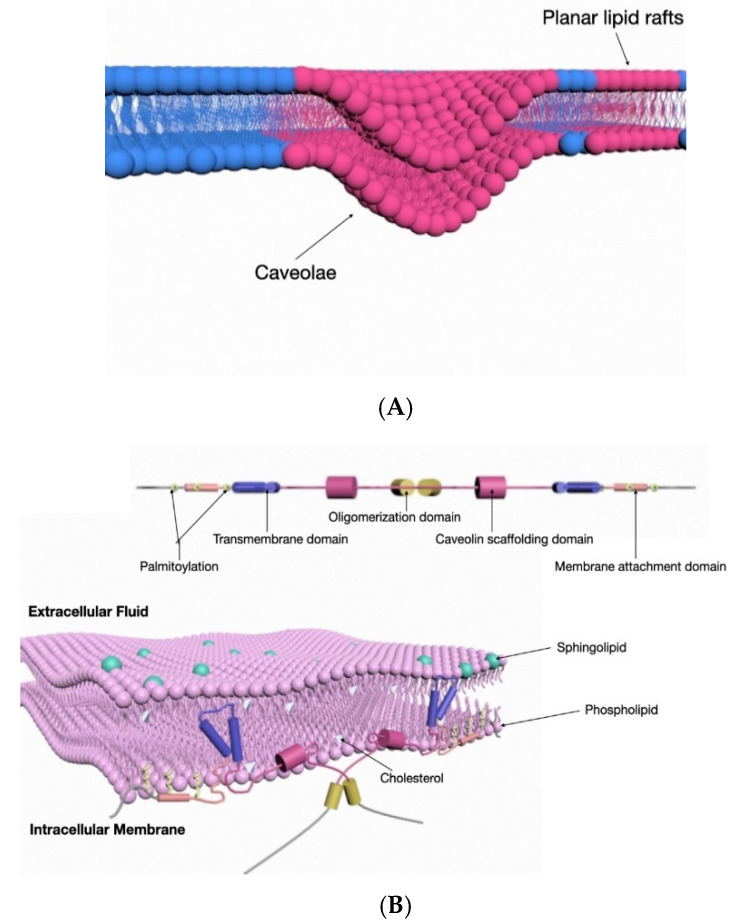
(**A**) Schematic illustration of caveolae and planar lipid rafts on the cell membranes. (**B**) Schematic diagram of caveolin-1 primary structure and predicted topology. Schematic showing that cav-1 is an integral membrane protein. The primary structure and predicted topology is illustrated here. Cav-1 often self-associates to a homo-oligomer which is called a caveolar assembly unit. Each caveolar assembly unit is composed of 14–16 monomers. Here, we show a dimer of cav-1 formed by two cav-1 monomers via the oligomerization domain (yellow color). The amino- and carboxyl-ends face towards the cytosol side. A hairpin loop (transmembrane domain, blue color) inserts into the membrane bilayer. The amino-terminal membrane-attachment domain is the caveolin scaffolding domain (CSD) (violet color).

**Figure 3 medsci-08-00046-f003:**
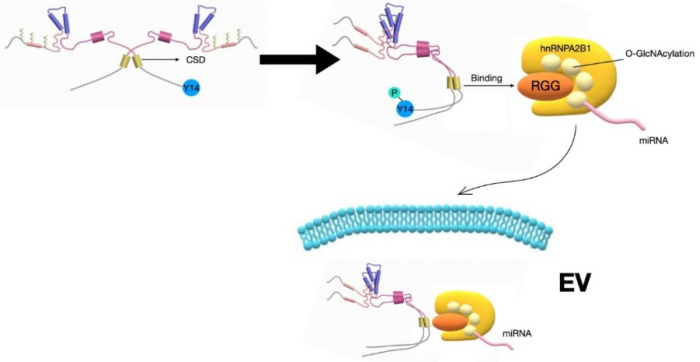
Schematic diagram of cav-1-mediated selection of EV miRNAs via hnRNPA2B1. In the presence of noxious stimuli, such as oxidative stress, cav-1 Y14 is phosphorylated and results in conformational change of cav-1. This pY14-induced change exposes the CSD to bind with the hnRNPA2B1 RGG domain. The interaction of cav-1 and hnRNPA2B1 leads to the encapsulation of both hnRNPA2B1 and its bound miRNAs into the EVs along with cav-1.

**Table 1 medsci-08-00046-t001:** List of well-documented markers for exosomes and microvesicles.

	Gene Name	Protein
Small EV markers	TSG101	Tumor susceptibility gene 101
	CD63	CD63 antigen
	TSPAN3	Tetraspanin-3
	TSPNA6	Tetraspanin-6
	ADAM10	Disintegrin and metalloproteinase domain-containing protein 10
Small-medium EV markers	HNRNPH1	Heterogeneous nuclear ribonucleoprotein H
	HNRNPL	heterogeneous nuclear ribonucleoprotein L
	VDAC1	voltage-dependent anion channel 1
	VDAC2	Voltage-dependent anion-selective channel protein 2
	PHB2	Prohibitin-2
	PDIA4	Protein disulfide-isomerase A4
	ATP5O	ATP synthase subunit O, mitochondrial
	SLC25A3	Phosphate carrier protein, mitochondrial
	RACGAP1	Rac GTPase activating protein 1
	KIF23	Kinesin-like protein KIF23

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
