# Peer review of "The Evolving Role of Caveolin-1: A Critical Regulator of Extracellular Vesicles"

_medsci, 2020, doi:10.3390/medsci8040046_

Round 1
Reviewer 1 Report
This is an excellently written, timely and comprehensive review on caveolin and EVs.
The authors have very comprehensively described using current literature the topics of caveolin and EVs. They have also highlighted the few publications identifying a role of caveolin in EV generation and uptake in parent cells as well as the role of Cav-1 in EV sorting.
I do have some minor comments that follow but also a general comment of the overall manuscript. Although there is substantial information on EVs (historical perspective etc) and Caveolin, the actual discussion of the role of Cav-1 and EVs is quite limited, which I appreciate is due to the lack of published data. For this reason, I would recommend reducing much of the heavy discussion at the beginning about EVs (history, nomenclature, biogenesis) as it is not required for the understanding of the discussion of the role of Cav-1 in EVs. Similarly, there should be a reduction in the lengthy discussion of Caveolins (different isoforms etc) as again this is not required for the understanding of the discussion of the role of Cav-1 in EVs. This information distracts from the overall message of the review which the reader has to wait until the very end to read. Removing much of the existing caveolin section to instead concentrate on the role of Cav-1 in endocytosis, exocytosis and signalling, which is only eluded to at Line 228-229, would provide a focussed background more in-line with the discussion of the role of Cav-1 in EVs. I appreciate this may take some time to rewrite but will make a more concise and informative review on the role of Cav1 in EVs (as the title suggests), as it currently reads as mini-reviews on EVs and caveolin independently.
Minor comments:
Q1 - Figure 1: why are there pictures of virus, bacteria and platelets? Is this to demonstrate size comparison? This needs to be made clear in the text and figure legend. It would appear in this figure that exosomes are produced by viruses etc. It is very unclear what the yellow image is meant to represent. Is the large grey image a cell and the smaller products the EVs, MVBs etc? These should look considerably different (different colour/texture?) or the cell should look different.
Q2 – Figure 1 legend: the figure does not demonstrate/illustrate how the EVs etc are processed as mentioned. Either include how they are processed or remove this sentence.
Q3 – Table 1: please keep nomenclature consistent. Either list EXO v MV v AB or Small EV v Large EV as discussed in lines 43-59 as some EXO would also appear in the sEV catergory. I would suggest keeping the nomenclature the same as in Figure 1.
Q4 – Line 72: Nomenclature has become quite confused in the text and not necessary in regard to my major comment (your discussion on cav1 in EVs). I would suggest removing most of this section and just tell the reader you will discuss the overall term EVs. Otherwise please correct the confusion in the text: Line 59 you mention small EV and large EV nomenclature is favourable and is official ISEV nomenclature, and that in line 37 exosome is mentioned as a historical error in naming. Now in line 72 you commence discussing the 3 categories of EXO, MV and AB. Again this doesn’t all fit with table 1.
Q5 – Line 97: what are production trains?
Q6 – Line 130-142: again the changing nomenclature (which I understand is due to how studies were published) is very confusing for the reader. If this manuscript will focus on EVs as mentioned, then I would suggest after figure 1 and table 1 to mention you will only focus on EVs and it will remove confusion and unnecessary information.
Q7 – Line 143 – 148: Lipid rafts are quite integral to this review. With the reduction of EV and Caveolin section content as recommended, inclusion of a small section introducing the reader to lipid rafts and the roles they play would be beneficial.
Caveolae are a specialised subset of lipid rafts. Lipid rafts are dynamic structures whose structure and composition is affected by temperature, environment, membrane dynamics etc whereas caveolae are specialised defined structures. This should be discussed at this line.
A figure comparing lipid rafts v caveolae would help the reader appreciate how specialised and different caveolae are to lipid rafts and also would also help the reader form connections between EVs, caveolin and caveolae.
Q8 – Line 164: This section can be removed and incorporated into the above section ending at line 163. The title does not deliver as little description is made on the expression of Cav in cells, instead focusing on Cav in tissue. It is mentioned that Cav1 is ubiquitous in all tissue but what about all cells? EV are very specific for their parent cell, so more comparisons with Cav and cells rather than tissues is really required.
Q9 – Line 166: Cav1 and caveolae are found in membranes of the noted structures (mitochondria etc), this should be mentioned. Caveolin is also predominantly expressed at the plasma membrane and should be noted as this is where the majority of caveolae are localised. See Line 184-186 where you discuss cav localisation from golgi to PM.
Q10 – Please correct to refer to Figure 2 not figure 1.
Q11 – Line 210 – 214: It would help to give an example of a residue and its interaction eg Y14 with Csk/Fyn. This residue is mentioned in Line 217-18 so could perhaps be reworked.
Q12 – repetition of sentences and references:
- Line 215 repeats content with Line 174-6
- Lines 226 – 8, both sentences are identical to those at lines 165-166 with identical referencing.
Q13 – Line 228 – 229: The role of cav1 in endocytosis, exocytosis and signalling is mentioned and referenced but not discussed further. The role of Cav1 in these cellular processes is integral to understanding the potential role of Cav1 in EVs. These points must be further discussed and referenced to current literature.
Author Response
Dear Editor,
We are pleased to submit a revised version of our manuscript medsci-921869 entitled "The Evolving Role of Caveolin-1: A Critical Regulator of Extracellular Vesicles" for consideration to be published in Medical Sciences. We have responded to the reviewer’s comments as detailed in the point by point responses below and addressed all the reviewers’ comments in both the rebuttal letter and the revised manuscript. We believe that the suggested revisions have strengthened the manuscript, and we hope that this revised manuscript will be suitable for publication in Medical Sciences.
Response to editor and reviewer: We appreciate the editor and reviewer’s comments. We revised our manuscript based on the new guidelines described above.
-------------------------------------------------------------------------------
Reviewer #1 (Comments to the Author (Required)):
Reviewer 1:
-I do have some minor comments that follow but also a general comment of the overall manuscript. Although there is substantial information on EVs (historical perspective etc) and Caveolin, the actual discussion of the role of Cav-1 and EVs is quite limited, which I appreciate is due to the lack of published data. For this reason, I would recommend reducing much of the heavy discussion at the beginning about EVs (history, nomenclature, biogenesis) as it is not required for the understanding of the discussion of the role of Cav-1 in EVs. Similarly, there should be a reduction in the lengthy discussion of Caveolins (different isoforms etc) as again this is not required for the understanding of the discussion of the role of Cav-1 in EVs. This information distracts from the overall message of the review which the reader has to wait until the very end to read.
Response to reviewer: We appreciate the reviewer’s comments, we have reduced the introductions on both EVs and cav-1 accordingly.
-Removing much of the existing caveolin section to instead concentrate on the role of Cav-1 in endocytosis, exocytosis and signalling, which is only eluded to at Line 228-229, would provide a focussed background more in-line with the discussion of the role of Cav-1 in EVs. I appreciate this may take some time to rewrite but will make a more concise and informative review on the role of Cav1 in EVs (as the title suggests), as it currently reads as mini-reviews on EVs and caveolin independently.
Response to reviewer: We appreciate the reviewer’s comments, and we have revised the manuscript and added more discussions on the role of cav-1 in endocytosis, exocytosis, and signaling. We kept some of the structural discussions as these structural motifs and modifications play key roles in the cav-1-regulated EV-cargo selection.
Minor comments:
Q1 to Q13: Response to reviewer: We appreciate the reviewer’s comments and help. We have revised the entire article and addressed each of these minor comments.
Q1 - Figure 1: why are there pictures of virus, bacteria and platelets? Is this to demonstrate size comparison? This needs to be made clear in the text and figure legend. It would appear in this figure that exosomes are produced by viruses etc. It is very unclear what the yellow image is meant to represent. Is the large grey image a cell and the smaller products the EVs, MVBs etc? These should look considerably different (different colour/texture?) or the cell should look different.
Response to reviewer: We appreciate the reviewer’s comments. We have edited the figure to make it clearer that the viruses, bacteria, and platelets are to demonstrate size comparison with small EVs, medium EVs, or large vesicles. We have also changed the color of vesicles to distinguish the vesicles from their mother cells.
Q2 – Figure 1 legend: the figure does not demonstrate/illustrate how the EVs etc are processed as mentioned. Either include how they are processed or remove this sentence.
Response to reviewer: We appreciate the reviewer’s comments. We have now edited the figure and revised the figure legend to demonstrate clearer the biogenesis of the different types of extracellular vesicles.
Q3 – Table 1: please keep nomenclature consistent. Either list EXO v MV v AB or Small EV v Large EV as discussed in lines 43-59 as some EXO would also appear in the sEV catergory. I would suggest keeping the nomenclature the same as in Figure 1.
Response to reviewer: We appreciate the reviewer’s comments and have now made the nomenclature easier to understand between the text, Figure 1, and Table 1.
Q4 – Line 72: Nomenclature has become quite confused in the text and not necessary in regard to my major comment (your discussion on cav1 in EVs). I would suggest removing most of this section and just tell the reader you will discuss the overall term EVs. Otherwise please correct the confusion in the text: Line 59 you mention small EV and large EV nomenclature is favourable and is official ISEV nomenclature, and that in line 37 exosome is mentioned as a historical error in naming. Now in line 72 you commence discussing the 3 categories of EXO, MV and AB. Again this doesn’t all fit with table 1.
Response to reviewer: We appreciate the reviewer’s comments and have now removed the section as suggested and addressed to the readers we will discuss the overall term EVs.
Q5 – Line 97: what are production trains?
Response to reviewer: We appreciate the reviewer for pointing this out. We have now changed the word choice.
Q6 – Line 130-142: again the changing nomenclature (which I understand is due to how studies were published) is very confusing for the reader. If this manuscript will focus on EVs as mentioned, then I would suggest after figure 1 and table 1 to mention you will only focus on EVs and it will remove confusion and unnecessary information.
Response to reviewer: We appreciate the reviewer’s comments. We now addressed in the paragraph after figure 1 and table 1 that we will focus on the overall term of EVs.
Q7 – Line 143 – 148: Lipid rafts are quite integral to this review. With the reduction of EV and Caveolin section content as recommended, inclusion of a small section introducing the reader to lipid rafts and the roles they play would be beneficial.
Caveolae are a specialised subset of lipid rafts. Lipid rafts are dynamic structures whose structure and composition is affected by temperature, environment, membrane dynamics etc whereas caveolae are specialised defined structures. This should be discussed at this line.
A figure comparing lipid rafts v caveolae would help the reader appreciate how specialised and different caveolae are to lipid rafts and also would also help the reader form connections between EVs, caveolin and caveolae.
Response to reviewer: We appreciate the reviewer for pointing this out. We have now included a section introducing lipid rafts to the readers. A schematic illustration demonstrating lipid rafts and caveolae is now added as a new panel in figure 2 (Figure 2A).
Q8 – Line 164: This section can be removed and incorporated into the above section ending at line 163. The title does not deliver as little description is made on the expression of Cav in cells, instead focusing on Cav in tissue. It is mentioned that Cav1 is ubiquitous in all tissue but what about all cells? EV are very specific for their parent cell, so more comparisons with Cav and cells rather than tissues is really required.
Response to reviewer: Thank you for the reviewer’s comments. As recommended, we have now removed the section and incorporated to the section above it.
Q9 – Line 166: Cav1 and caveolae are found in membranes of the noted structures (mitochondria etc), this should be mentioned. Caveolin is also predominantly expressed at the plasma membrane and should be noted as this is where the majority of caveolae are localised. See Line 184-186 where you discuss cav localisation from golgi to PM.
Response to reviewer: We appreciate the reviewer’s comments and have now stated that caveolae can be found in the membranes of the structures mentioned in the text as well as on the plasma membrane.
Q10 – Please correct to refer to Figure 2 not figure 1.
Response to reviewer: We appreciate the reviewer for pointing this out and have made the correction.
Q11 – Line 210 – 214: It would help to give an example of a residue and its interaction eg Y14 with Csk/Fyn. This residue is mentioned in Line 217-18 so could perhaps be reworked.
Response to reviewer: We appreciate the reviewer’s comments. However, we have not found the literature showing that Y14 interactions with Csk/Fyn regulate EV production or EV-cargo selection. We therefore did not quote this example. We indeed had described examples showing interactions between CSD Y99 and HnRNPA2B1, and how this interaction regulates EV-cargo miRNA selections.
Q12 – repetition of sentences and references:
- Line 215 repeats content with Line 174-6
- Lines 226 – 8, both sentences are identical to those at lines 165-166 with identical referencing.
Response to reviewer: We appreciate the reviewer for pointing this out and have now removed those repetition.
Q13 – Line 228 – 229: The role of cav1 in endocytosis, exocytosis and signalling is mentioned and referenced but not discussed further. The role of Cav1 in these cellular processes is integral to understanding the potential role of Cav1 in EVs. These points must be further discussed and referenced to current literature.
Response to reviewer: We appreciate the reviewer’s comments. We have made edits in this section and throughout the article, while keeping the focus of the section on discussing caveiolin-1’s expression in EVs and their role in EV generation and EV uptake.
Yang Jin, MD., PhD
Professor of medicine.
Pulmonary and Critical Care Medicine
Boston University Medical Campus
72 E. Concord St., Boston, MA 02118
Reviewer 2 Report
The article is interesting and well structured, however a minor revision should be done.
- the paragraph "caveoli-1 expression in different cells" should be eliminated by moving the description of the two isoforms cav1-alpha and beta in the previous paragraph.
- the paragraph "caveolin-1 structure and isoforms" should be abbreviated as the detailed description of the caveolin structure, except for the phosphorylation sites, it is not useful to illustrate its regulatory role for the release / formation of extracellular vesicles.
Author Response
Dear Editor,
We are pleased to submit a revised version of our manuscript medsci-921869 entitled "The Evolving Role of Caveolin-1: A Critical Regulator of Extracellular Vesicles" for consideration to be published in Medical Sciences. We have responded to the reviewer’s comments as detailed in the point by point responses below and addressed all the reviewers’ comments in both the rebuttal letter and the revised manuscript. We believe that the suggested revisions have strengthened the manuscript, and we hope that this revised manuscript will be suitable for publication in Medical Sciences.
-The article is interesting and well structured, however a minor revision should be done. the paragraph "caveoli-1 expression in different cells" should be eliminated by moving the description of the two isoforms cav1-alpha and beta in the previous paragraph.
Response to reviewer: We appreciate the reviewer’s comments. We have revised the manuscript accordingly.
-The paragraph "caveolin-1 structure and isoforms" should be abbreviated as the detailed description of the caveolin structure, except for the phosphorylation sites, it is not useful to illustrate its regulatory role for the release / formation of extracellular vesicles.
Response to reviewer: We appreciate the reviewer’s comments. We have revised the manuscript accordingly. However, majority of the structure of cav-1 is important for readers to understand how cav-1 regulates EV-cargo selection. For example, cav-1 phosphorylation on Y14 is a key point mediating interaction with RNA-holding protein HnRNPA2B1 and subsequently guiding them into EVs. Therefore, we have to leave the majority of the cav-1 structure description in the text.
Yang Jin, MD., PhD
Professor of medicine.
Pulmonary and Critical Care Medicine
Boston University Medical Campus
72 E. Concord St., Boston, MA 02118
Round 2
Reviewer 1 Report
The review reads excellently and is a very comprehensive study of caveolin, extracellular vesicles, and the newly identified role of Cav in EV biology,
Author Response
We appreciate the editor and reviewer’s comments
Reviewer 2 Report
The authors reviewed and organized the text by evaluating the comments made previously. The manuscript is now suitable for publication.
Author Response

(The authors gave the same response as above.)
